# Twisting of Fibers Balancing the Gel–Sol Transition in Cellulose Aqueous Suspensions

**DOI:** 10.3390/polym11050873

**Published:** 2019-05-13

**Authors:** Dmitry V. Zlenko, Sergey N. Nikolsky, Alexander S. Vedenkin, Galina G. Politenkova, Aleksey A. Skoblin, Valery P. Melnikov, Maria G. Mikhaleva, Sergey V. Stovbun

**Affiliations:** 1Faculty of Biology, M.V. Lomonosov Moscow State University, Lenin Hills 1/12, 119192 Moscow, Russia; 2N.N. Semenov Institute of Chemical Physics, RAS. Kosygina 4, 119991 Moscow, Russia; nikolskij56@mail.ru (S.N.N.); a.s.vedenkin@gmail.com (A.S.V.); g_politenkova@mail.ru (G.G.P.); ab1954@yandex.ru (A.A.S.); melnikov@chph.ras.ru (V.P.M.); wawe@bk.ru (M.G.M.); s.stovbun@yandex.ru (S.V.S.)

**Keywords:** cellulose hydrogel, cellulose films, twisting, dispersing

## Abstract

Cellulose hydrogels and films are advantageous materials that are applied in modern industry and medicine. Cellulose hydrogels have a stable scaffold and never form films upon drying, while viscous cellulose hydrosols are liquids that could be used for film production. So, stabilizing either a gel or sol state in cellulose suspensions is a worthwhile challenge, significant for the practical applications. However, there is no theory describing the cellulose fibers’ behavior and processes underlying cellulose-gel-scaffold stabilizing. In this work, we provide a phenomenological mechanism explaining the transition between the stable-gel and shapeless-sol states in a cellulose suspension. We suppose that cellulose macromolecules and nanofibrils under strong dispersing treatment (such as sonication) partially untwist and dissociate, and then reassemble in a 3D scaffold having the individual elements twisted in the nodes. The latter leads to an exponential increase in friction forces between the fibers and to the corresponding fastening of the scaffold. We confirm our theory by the data on the circular dichroism of the cellulose suspensions, as well as by the direct scanning electron microscope (SEM) observations and theoretical assessments.

## 1. Introduction

Cellulose is a common biological polymer, widely used in industry due to its excellent mechanical properties, biodegradability, and comparatively low cost. Natural cellulose (and chitin) consists of macroscopic fibers composed of smaller and mechanically stronger helical elements [1,2,3]. The smallest building block of the natural cellulose is an elementary fibril or nanofibril, which is a twisted bundle of parallel cellulose chains [2,4,5]. The lateral size of the nanofibril depends on the origin of the cellulose, and is in the range of 3–4 nm in softwood [6,7,8,9], to 9 nm in cotton [10], and up to 20 nm in cellulose from tunicates [5]. Nanofibrils twist into microfibrils forming helical bundles [11,12,13]. The hierarchical supramolecular organization of cellulose provides for its mechanical properties in nature [1,14].

The enzymatic hydrolysis of cellulose, followed by ultrafine mechanical milling in water, leads to forming hydrogels that contain 1%–3% cellulose by mass [15]. The same result could be achieved through aggressive acid hydrolysis followed by sonication [16]. The lattice of cellulose hydrogels is made up of nanofibers (nanocellulose) having a rather high aspect ratio, enriched with a crystalline fraction [17] and demonstrating a prominent helical structure [8,18]. Increasing hydrolysis time or intensifying the dispersion process leads to a decrease in both the length of the nanofibers and their polydispersion [2,18].

Once the concentrated aqueous solutions of nanocellulose are dried, dense films may form [2,19,20]. Nanocellulose films often demonstrate unique optical properties due to the chiral nematic phase formation [2,18,20,21]. Water evaporation decreases the cellulose nanofibers’ helix pitch, which, in its turn, results in coloration [2,19,22]. The latter is related to the supramolecular packing of nanofibrils within the film, and strongly depends on the heterogeneity of the fibrils. Thus, the coloration effect is tied to the protocol of the raw-material treatment [19]. Although many works focused on adjusting nanocellulose properties, the structure–property relationship between nanocellulose fibers and their chiral phases and gels is still relatively unstudied [2,8]. For example, increasing cellulose nanofibers’ surface-bound charge can increase the pitch of the chiral phase in solution, and, thus, promote the liquid crystalline phase formation [23]. Meanwhile, the density of the cellulose films is also related with the charge density on the fibers’ surface, while the mechanical properties of the films correlate mostly with the crystallinity index [17].

Cellulose, chitin, and chitosan hydrogels are themselves advantageous materials, applied in many fields of medicine and agriculture [24,25,26]. For example, cellulose-based hydrogels can be used as water/nutrient reservoirs [24], for stomach bulking, for drug delivery [27], or for wound dressing [27,28]. However, cellulose films obtained by the hydrogels’ drying are also of particular interest, for example, as biodegradable and safe material for food packing [29].

The properties of nanocellulose suspensions critically depend on the preparation conditions and cellulose’s origin [2,8,16,17,30]. In some cases, the hydrogel properties can significantly restrict the formation of the dense films. This phenomenon was described as the “nonequilibrium kinetic arrest” of the gel-like state in the cellulose suspension preventing the formation of dense films upon water evaporation [23,31]. Yet, in case the gel-like state was not “arrested”, suspension become unsuitable for stable hydrogel production. So, the problem to stabilize one of these states in nanocellulose suspensions is of crucial importance for modern high-end industrial applications. Nevertheless, there is no universal theoretical description of the processes stabilizing the 3D scaffold of nanocellulose gels.

The cellulose hydrogel formation and stabilization are usually explained through van der Waals interactions and hydrogen bonding between nanocellulose fibers [25,30,32,33]. However, in natural cellulose, hydrogen bonds predominantly form between the macromolecules rather than between the nanofibrils [24,34,35]. The spatial complementarity of cellulose chains is essential for short-range hydrogen-bond formation, and this condition seems to not be satisfied in case of an irregular gel lattice. Still, nanocellulose forms stable hydrogels [23,24].

In this work, we demonstrate that the extent to which cellulose is ground determines the stability of the resulting hydrogel. Mild grinding and dispersing results in less stable hydogels that could be used to produce dense and transparent films via simple drying. On the other hand, intensive dispersing (such as sonication) leads to the gel-like hydrogels that form white, opaque, and shapeless dense pieces upon drying. Such hydrogels could not be used to produce films, but are stable enough to be utilized as hydrogels themselves [24]. Based on our previous works, we provide a phenomenological description of the mechanism explaining the behavior of the nanocellulose suspensions and its dependence on the degree of the milling. The main idea is that, in case of intense dispersing, cellulose was able to reassemble into the morphological structures resembling natural supramolecular packing. This leads to stabilizing the hydrogel scaffold in 3D form and preventing film formation.

## 2. Materials and Methods

The hardwood bleached craft cellulose (GOST 28172-89, Figure 1A) was obtained from Arkhangelsk pulp-and-paper mill (Russia). The cellulose suspensions were prepared in 3 steps: 1. oxidative hydrolysis; 2. preliminary grinding; 3. final dispersing. Acid hydrolysis or oxidative acid hydrolysis is a standard initial step for cellulose hydrogel preparation [2,23,36,37]. The subsequent treatment could be mechanical grinding [8,15] or sonication [2,16,23]. Here, we used both approaches.

Cellulose hydrolysis (2 h) was held in the mixed aqueous solution of sulfuric acid (10%) and hydrogen peroxide (3%) at a temperature of 95–$97{\;}^{\circ }$C and a solvent-to-pulp ratio of 30 (per mass). After cooling the mixture, the reaction was stopped by flushing with cold distilled water on a glass filter. Then, the pulp was drained with cold distilled water until pH reached 7.0. Then, the suspension of cellulose (2.5%–2.8% per mass) was ground by a colloidal mill MK 2000 (IKA, Germany) at the room temperature. The rotor–stator clearance was smaller than 10μm, and milling time was 30 min (Figure 1C). The obtained cellulose pulp (2.5%–2.8%) was diluted 10 times with distilled water and dispersed using high-pressure homogenizer HPH 2000/4-DH5 (IKA, Germany), at pressure of 1.8–1.9 kBar. The procedure was repeated 10 times. The obtained cellulose suspensions (∼2.5 mg/mL, Figure 1D) were white viscous liquids (Figure 1E), so we refer to them as “hydrosols”. Hydrosols were diluted twofold with distilled water and sonicated using immersible source UZG 8–0.4/22 (22 kHz, 0.4 kW, VNIITVCH, Sent-Petersburg, Russia) for 5 min. The resulting suspensions (∼1 mg/mL) were jellylike nonflowing white substances (Figure 1F) as was previously shown [29]. So, here and hereafter this type of the cellulose suspension is referred to as “hydgels”. The grinding procedures were independently repeated 5 times.

The morphology of the cellulose in suspensions was examined microscopically in the xerogel and aerogel samples. Xerogel samples were prepared as follows: the samples of the cellulose hydrosol were diluted 100 times (down to ∼25 ng/mL) with the distilled water and incubated under stirring for 1 hour. The aliquots of the obtained material were dripped on the glass slide and dried at room temperature for 3 h. Prior to the experiments, the glass surface was sonicated and then washed with 100% ethanol. The aerogel samples [25,33] were prepared as follows: the cellulose hydrogel droplets (2–4 mm in diameter) were frozen in liquid nitrogen and lyophilized at $-25{\;}^{\circ }$C.

The samples of the cellulose suspensions in the course of preliminary treatment and milling (Figure 1A–D) were investigated using a Phenom XL (Phenom World) scanning electron microscope. After each stage of treatment, the samples of the cellulose pulp were placed on the glass slide, dried at $70{\;}^{\circ }$C for 12 h, and examined. The xerogel samples were examined using a Solver UHV (NT-MDT, Russia) atomic force microscope (AFM) operating in a semicontact mode. Semicontact mode is preferable for the soft and easily destroyable materials, and it allows for achieving a resolution of 1 nm. Aerogel samples were investigated by a JSM-7500F (Jeol, Japan) scanning electron microscope (SEM) operating at low accelerating voltage (0.5–1.5 kV). Aerogel samples were not covered by metal.

The twisting/untwisting of the cellulose fibers was monitored by circular-dichroism (CD) spectra. CD spectra were measured using an SKD-2 CD spectrometer [38]. Cellulose films were prepared from liquid mass by drying at the glass slide at room temperature. The chemicals were obtained from ChimMed (Moscow, Russia).

## 3. Results and Discussion

### 3.1. Hydrosol Sedimentation Stability and Structure

The problem to find the right equilibrium between stabilizing and destabilizing the cellulose gel structure is important for practical applications [8,16,23,30,31], as cellulose can be used in the form of the hydrogel itself [25,39] or in the form of the film [2,24].

At the same time, even the simple sedimental stability of the cellulose gels was not previously estimated, so we believe this task is noteworthy.

To verify the cellulose suspensions’ long-term stability, 12 independent samples of hydrosol (1.1%, 1.4%, 1.8%, and 2.7%, by mass, three repeats per each concentration) were incubated at room temperature for two years. During this period, we did not observe any precipitates or optical inhomogeneities, irrespective of gel concentration. So, in the hydrosol, the characteristic time of cellulose precipitation or aggregation lies in a yearly time-scale.

The AFM investigation of dried hydrosols revealed chaos of the short (<1 μm on average) and rather thick (∼100 nm on average) fibers. Let us estimate the characteristic time of precipitation of such fibers. The velocity (*v*) of the sedimentation of spherical particles having a diameter *d* could be estimated according to the expression:$$v=\frac{g\left(\rho -\rho_{0}\right)d^{2}}{18\eta }$$
where ρ and $\rho_{0}$ are the density of the particles and surrounding medium, correspondingly; *g*, gravity acceleration; and η, viscosity of the medium. The same estimate can be considered as an upper-bound assessment of the sedimentation rate of the cylinder-shaped particles of the same diameter (*d*). So, sedimentation of the micron-sized cellulose hydrosol elements should become noticeable in a time scale of days, while hydrosols were stable for at least two years.

The AFM investigation of the hydrosols also revealed elements of a smaller diameter of about 100 nm (Figure 2B). These particles should also precipitate in the yearly time-scale. However, sedimentation was not observed, which implies that cellulose fibers were interacting with each other, forming a continuous lattice, which accounts for the sedimental stability of the hydrosol.

Let us estimate the energy of interaction between cellulose fibers in hydrosol. The energy (*W*) of interaction between two parallel cellulose fibers having radius R∼100 nm and length L∼1μm could be estimated as [40]:$$W=\frac{AL\sqrt{R}}{24\sqrt{h^{3}}},$$
where *h* is the distance between the fibers, and $A=10^{-20}$ J – Hamaker constant [40]. The Coulomb friction force ($F_{f}$) is related to downforce $F_{d}$ and a friction coefficient ($k_{f}$):$$F_{f}=k_{f}F_{d}$$
The downforce could be calculated as a partial derivative of the interaction energy between the fibrils per the distance between them:$$F_{d}=-\frac{\partial W}{\partial h}=\frac{AL\sqrt{R}}{16\sqrt{h^{5}}}$$
Assuming $k_{f}∼1$, and the distance between the fibers ∼ nm (Figure 2), force $F_{f}$ would be about $10^{-9}$ N. For the crossed fibrils, the friction force would be smaller in proportion to the ratio of the fiber length and diameter, i.e., approximately $∼10^{-10}$ N.

The gravity force acting on the cellulose fiber and responsible for sedimentation could be assessed as follows (regardless the Archimedes buoyant force):$$F_{G}=mg=V\rho g=\pi R^{2}L\rho g∼10^{-16}\;N$$
where the density on cellulose ρ was assumed to be ∼1.5 g/mL. Therefore, the gravity force is many orders of magnitude smaller than the friction force between the crossed and tightly clamped-down cellulose fibers. So, hydrosol’s stability could be explained by van der Waals interaction between the fibrils.

### 3.2. Hydrogel Structure

The morphology of the cellulose hydrogels was examined in the aerogel samples. The lyophilization of gel droplets frozen at 77 K preserves the original structure of the gel lattice [25,33] that can then be visualized using SEM (Figure 3). The scaffold of the cellulose hydrogel looks like a continuous irregular net of the helical cellulose fibers with a diameter of about 10–20 nm, occasionally united in the thicker fibers, up to 100 nm in diameter. On average, the length of the thin fibers was about several hundred nanometers. Similar dimensions were earlier reported for cellulose hydrogels treated in a similar manner [15,18,31].

The helical structure and dimensions of the cellulose fibrils in the hydrogel strongly resemble the hierarchical structure of the native cellulose [1,3]. The fibers of several dozen nanometers in diameter correspond to native cellulose microfibrils usually having a diameter of about 15–30 nm [12,24]. Thicker fibers could be considered as the bundles of the microfibrils that were also observed in plant cell walls [12,24,35]. The relatively small length of the fibers (∼1 μm) in the hydrogel implies the breaking of the microfibrils in the course of the milling and homogenization (Figure 1B).

The congruence of elements’ sizes in the hydrogel and natural cellulose most probably points at preserving the native cellulose structure despite of the acid hydrolysis, grinding, and sonication. The main structural difference between native cellulose and the hydrogel’s scaffold is that fibrils composing the hydogel wind around each other and form a 3D net (Figure 3, arrows) instead of the regular layered structure typical of the plant cell wall [6,41,42]. Similar images were demonstrated earlier for the sonicated cellulose hydrogels [2,5,18]. This observation implies that, in the hydrogel’s scaffold, the cellulose fibrils have, in part, preserved their native structure and, in part, have formed some new contacts resembling the native ones. The latter were randomly formed and accounted for stabilizing the hydrogel’s scaffold. The friction force between the twisted fibers increases exponentially according to the Capstan equation:$$F_{f}∼e^{2\pi \alpha k_{f}}$$
where α is the number of turns. This effect is absent in a more-or-less liquid hydrosol, and makes the hydrogel a jellylike substance capable of shape-keeping.

### 3.3. Cellulose Films

Fine cellulose suspensions are known to form dense films upon drying, but the properties of the obtained suspensions and films strongly depend on the treatment protocol [2,18,31,43]. For example, the films’ coloration depends on the size distribution of the cellulose nanocrystals [19], while some treatment conditions (acid hydrolysis time [44]), as well as the system composition [23] may lock the suspension in a gel-like state.

We attempted to prepare the films using a cellulose suspension taken at different stages of treatment (Figure 1). Upon drying, the cellulose only pretreated with a colloidal mill (Figure 1C) formed white, opaque, and spongy films (Figure 4, top). On the other hand, after the high-pressure treatment, the obtained hydrosols formed dense, transparent, and colorless films (Figure 4, bottom). The latter films were significantly thinner.

The difference in behavior of the milled and additionally homogenized cellulose could be explained by the crucial difference in the dimension of the elements present in the mixture (Figure 1C,D). The suspension treated with a colloidal mill still contained the intact fragments of plant cells that are absent in the hydrosol. Too-large and hollow fragments form a cavernulous and highly scattering film (Figure 4, top). On the other hand, the hydrosol was composed of much smaller elements that were able to agglomerate in a dense and uniform film.

It would be reasonable to propose that the sonicated cellulose suspension (hydrogel) would form denser films than the hydrosol, as the additional dispersing treatment would decrease the average size of the cellulose fragments and increase their ability to agglutinate. However, upon drying, the pieces of hydrogel formed irregular-shaped opaque blocks instead of films. It seems that there is a clear analogy between the observed effect and the earlier-reported “kinetic arrest” of the gel-like state described for concentrated cellulose suspensions [23]. Note that, in our experiments, the hydrogel samples were diluted twice, as compared to the hydrosol, so, the concentration cannot explain the gel stabilization.

Sonication is a widely used approach for dispersing cellulose and preparing hydrogels [2,16,23]. However, our results demonstrate that if the final goal of the work is film production, so sonication seems to be too strong, thus promoting locking the suspension in a gel state.

### 3.4. Optical Activity of Cellulose Hydrosol

Based on SEM observations, we proposed that gel-state stabilization after sonication occurs due to the occasional mutual twisting of the cellulose fibrils that leads to forming the continuous 3D lattice. Twisting cellulose fibrils into novel structures should be precluded by a partial untwisting of the existed structures [3]. The latter can be monitored through measuring the optical activity (circular dichroism spectra) of the samples, as the contribution of twisting the crystalline nanofibrils seems to be greater than the optical activity of individual glucose monomers [38,45]. So, variations in the CD spectra observed under nondestructive conditions clearly indicate the twisting/untwisting processes in cellulose.

The cellulose hydrosols demonstrated prominent optical activity in the near-UV region (Figure 5A). The maximum of the CD spectrum shifted hypsochromically with the decrease of cellulose concentration (Figure 5A, inset). This effect can be explained by the cellulose fibers untwisting caused by dilution, as untwisting leads to increasing the void volume in the samples and decreasing the density of the fibers’ packing, correspondingly. The latter leads to widening the band gap and the corresponding shift of the absorption maximum, which directly affects the CD maximum position. The UV absorption spectra themselves were crucially corrupted by light scattering in the turbid cellulose hydrosol samples (data not shown). So, the observed hypsochromic shift could be considered as an indication of an at least partial untwisting of the nanofibrils.

The untwisting o cellulose fibers should be intensified by the electrostatic repulsion of the nanofibrils in case of charging their surface [17,23]. For example, the sulphiting of the cellulose (in the course of the sulfite pulping process) should result in negatively charged nanofibrils. Charging the nanofibrils’ surface was reported even for a much-stronger sulphuric acid [23]. Treating hydrosol samples with a sodium hydrosulphite affected the CD spectra in accordance with our assumption (Figure 5B). The amplitude of the CD signal decreased with increasing NaHSO${}_{3}$ concentration (Figure 5B, inset). So, the twisting–untwisting processes take place in cellulose suspensions upon some external stimuli. SEM observations and data on the optical activity, taken together, brought us to the conclusion that stabilization of the cellulose hydrogel structure can occur due to the mutual twisting of fibers detached from each other by sonication.

## 4. Conclusions

Cellulose nanofibrils can stick together to form a 3D net that serves as the gel scaffold [25,39], or to form the dense and transparent or even colored films [2,31]. Here, we demonstrated the crucial influence of cellulose treatment intensity on the properties of the final cellulose suspensions and films. In particular, sonication seems to be strong enough to cause partial cellulose untwisting, which leads to reverse twisting after the ultrasound was switched off. The twisting back can occur randomly in the bulk of the suspension that stabilizes the 3D scaffold of the hydrogel and prevents dense-film formation.

On the other hand, the mechanical strength of individual nanofibrils is much greater than the strength of macroscopic cellulose fibers [46]. The mutual arrangement of the cellulose nanofibrils in the natural materials is optimized for their functions in the organism, but not for industrial applications. Rearrangement of the natural nanofibrils’ packing is a very promising way to produce the novel materials. For example, flow-assisted orientation and agglomeration of the nanocellulose fibers allowed to create a fiber with tensile strength up to 1.5 GPa [36]. Considering our hypothesis on the role of the twisting in the hydrogels’ scaffold stabilization and its role in other processes in cellulose pulp, we anticipate that the next step in the design of cellulose-based materials would be controlled aggregation and twisting of the individual macromolecules into a thick fiber (“huge nanofibril”) having overwhelming mechanical properties.

## Figures and Tables

**Figure 1 polymers-11-00873-f001:**
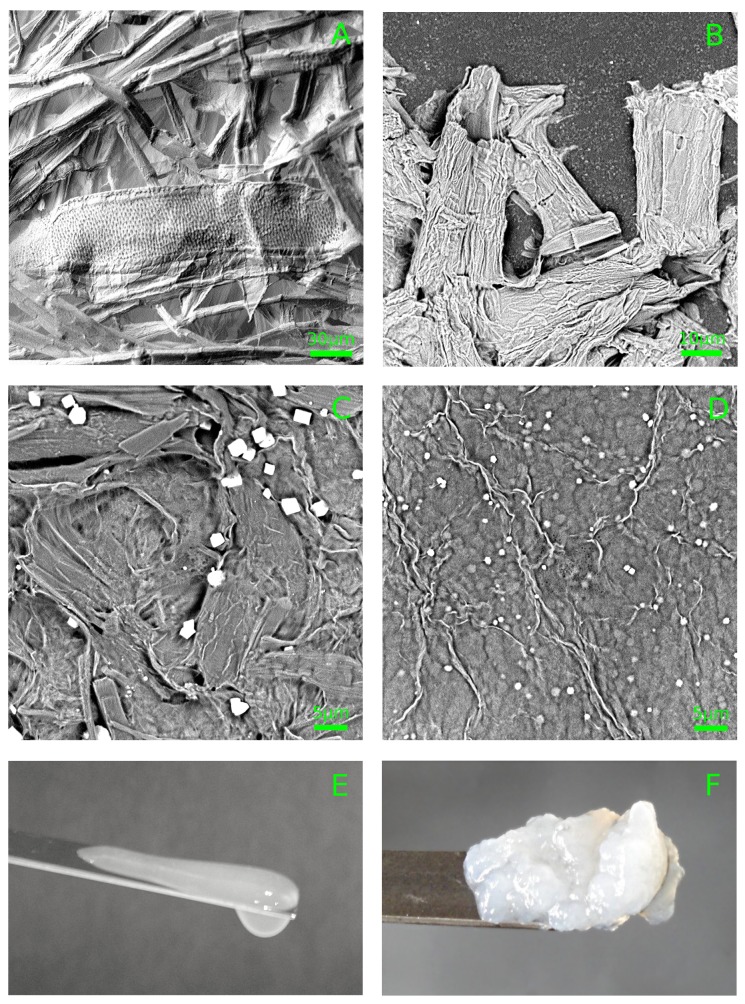
(**A**–**D**) Typical scanning electron microscope (SEM) images of cellulose pulp in the course of milling. (**A**) Initial raw; (**B**) pulp after oxidative hydrolysis; (**C**) pulp treated with a colloidal mill; (**D**) cellulose after final dispersing with a high-pressure homogenizer (hydrosol). White blocks correspond to salt crystals. Pulp after high-pressure homogenization (hydrosol) was a viscous liquid (**E**) that became a jellylike substance (hydrogel) after sonication (**F**).

**Figure 2 polymers-11-00873-f002:**
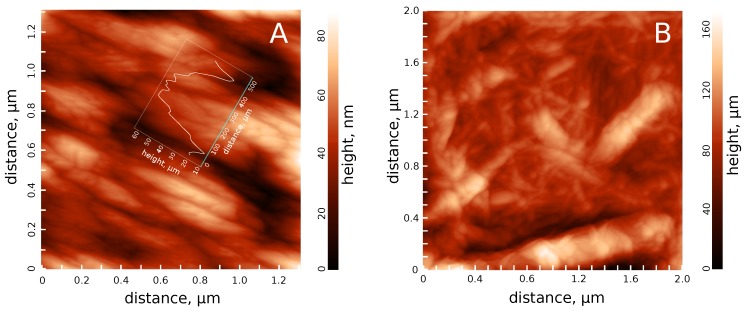
Typical atomic force microscope (AFM) images of the dried cellulose hydrosol. Cellulose was passed through the high-pressure homogenizer but was not sonicated. Some regions were composed of the more-or-less parallel elements (**A**), while in others, the cellulose elements crossed (**B**). The inset in (**A**) shows the horizontal relief profile.

**Figure 3 polymers-11-00873-f003:**
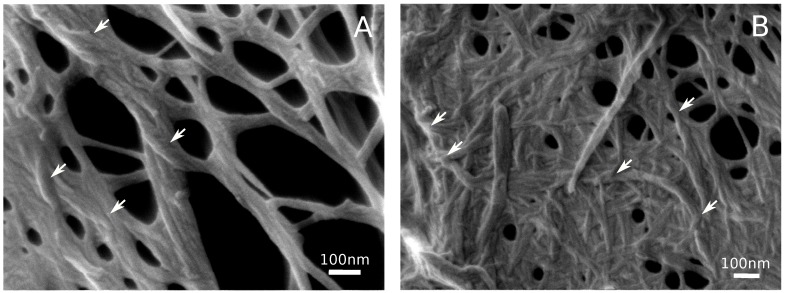
Typical SEM images of the lyophilized nanocellulose hydrogels (aerogels). Aerogel samples were rather heterogeneous, some areas were sparse (**A**), while others were dense (**B**). Arrows indicate helical and twisted zones of contact between fibrils.

**Figure 4 polymers-11-00873-f004:**
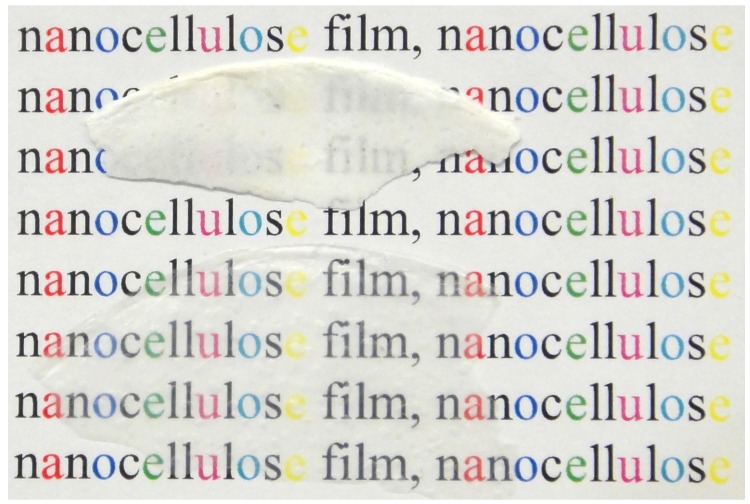
Cellulose films obtained from the pulp only treated with a colloidal mill (top, Figure 1C), and additionally treated with the high-pressure homogenizer (bottom, Figure 1D). Hydrogels obtained after sonication of the cellulose suspension did not form films upon drying.

**Figure 5 polymers-11-00873-f005:**
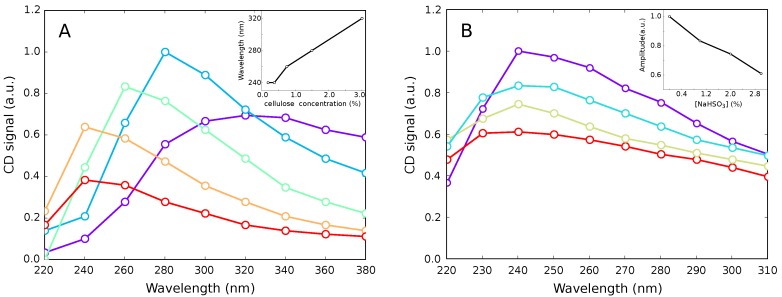
Circular-dichroism spectra of the cellulose hydrosols of different concentration (**A**), and in the presence on sodium hydrosulfite (**B**). (**A**) purple—3.0% of cellulose by weight, blue—1.5%, green—0.75%, orange—0.38%, and red—0.19%. Inset: position of the maximum as a function of cellulose concentration. (**B**) violet—pure cellulose hydrosol, cyan—1.0% of NaHSO${}_{3}$ by weight, yellow—2.0%, and red—3.0% (cellulose concentration, 0.38%). Inset: spectra maximum amplitude as a function of NaHSO${}_{3}$ concentration.

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
