# Peer review of "Twisting of Fibers Balancing the Gel–Sol Transition in Cellulose Aqueous Suspensions"

_polymers, 2019, doi:10.3390/polym11050873_

Reviewer 1 Report

Comments to the author: Reject

In this work, the authors discussed the nanocellulose hydrogel stability and sedimentation, which should be a very interesting work. However, this paper also gave some other results, for example, cellulose films and the optical activity of hydrogel, it make me confused. In my opinion, we should focus on what factors may have effects on the stability and sedimentation of hydrogel, such as pH value, shear force, temperature, etc. I did not see enough valuable information for this work, so I think this work can not be published in Polymers.

 Some comments are listed as follow:

 1. English should be significantly improved.

 2. Line 78. What is the meaning of “a hydromodulus of 30”?

 3. Line 81. Check “The hydrolyzed cellulose raw”, very strange description.

 4. Figure 2. Please note the A and B

 5. Line 361. “scanning EM” should be “SEM”

 6. Line 293. What is the aim of measuring the optical activity of hydrogel? 

Author Response

@page { margin: 2cm } p { margin-bottom: 0.25cm; line-height: 120% }

Dear Reviewer #1,

Thank you for the critical review and notes.

In this work, the authors discussed the nanocellulose hydrogel stability and sedimentation, which should be a very interesting work. However, this paper also gave some other results, for example, cellulose films and the optical activity of hydrogel, it make me confused. In my opinion, we should focus on what factors may have effects on the stability and sedimentation of hydrogel, such as pH value, shear force, temperature, etc. I did not see enough valuable information for this work, so I think this work can not be published in Polymers.

– We have carefully revised the Manuscript and corrected the title which was not suitable for the presented results. The altered title sound like this: “The Influence of the Dispersing Intensity on the Gel-Sol Transition in Aqueous Cellulose Suspensions” We also reduced the amount of the material to clarify the Manuscript.

Some comments are listed as follow:

1. English should be significantly improved.

– We have revised the text and proofread the English

2. Line 78. What is the meaning of “a hydromodulus of 30”?

– Sorry about that. It was a loan translation from Russian. The phrase means: “The solvent to pulp ratio was 30”

3. Line 81. Check “The hydrolyzed cellulose raw”, very strange description.

– The sentence was revised along with the whole text.

4. Figure 2. Please note the A and B

– We have provided two images to show the variability of the morphology of the obtained xerogels. Now, this fact is outlined in the capture to Figure 2.

5. Line 361. “scanning EM” should be “SEM”

– The sentence was corrected.

6. Line 293. What is the aim of measuring the optical activity of hydrogel?

– The optical activity may appear due to the molecular chirality of substance, as well as due to the chiral supramolecular ordering. The provided CD spectra demonstrate that the twisting/untwisting processes take place in the cellulose suspensions that is important for our reasoning. We have revised and appended the corresponding section of the Manuscript to make it clearer.

Reviewer 2 Report

The manuscript entitled "Nanocellulose Hydrogels Long-Term Stability and Sedimentation" describes a phenomenological description of a mechanism that could potentially explain a transition between the stable gel and shapeless liquid states. Even if I think that the topic is not entirely novel, the paper is well written and the main approach could be interesting for readers. Therefore, I think that it is suitable for the publication in the Polymers Journal. However, there are important points to be considered as the following:

- The text in the results Section seems to contain information that should be explained in the methods Section, such as equations, while results should provide some additional numerical information that will support the study.

- The sentence 118: "... cellulose hydrogel (not sonicated) were stable for years." is not entirely proved.

- Paragraph 141 - 150 is very speculative. Please, provide references or data/information that supports authors claim.

- The conclusions Section is a mix of discussion and conclusions. Please clarify these paragraph and provide clear conclusions of this study. References should be avoided to highlight just only the authors´findings. 

- This whole paper needs to be submitted to English language review by a native speaker or a manuscript editing service.

Author Response

@page { margin: 2cm } p { margin-bottom: 0.25cm; line-height: 120% }

 Dear reviewer #2,

Thank you for the positive review and the notes.

Comments and Suggestions for Authors

The manuscript entitled "Nanocellulose Hydrogels Long-Term Stability and Sedimentation" describes a phenomenological description of a mechanism that could potentially explain a transition between the stable gel and shapeless liquid states. Even if I think that the topic is not entirely novel, the paper is well written and the main approach could be interesting for readers. Therefore, I think that it is suitable for the publication in the Polymers Journal. However, there are important points to be considered as the following:

- The text in the results Section seems to contain information that should be explained in the methods Section, such as equations, while results should provide some additional numerical information that will support the study.

Sorry, but we do not agree with you on this point. The provided equations and assessments cannot be considered as common and well known. These equations more likely represent the physical model rather than the methodology. So, we need to discuss them along with the numerical data presentation.

- The sentence 118: "... cellulose hydrogel (not sonicated) were stable for years." is not entirely proved.

We have appended the Methods section to clarify this point. Twelve independent samples were incubated for a bit longer than two years, and nothing happened with them. Nevertheless, we have removed this too straight statement from the text.

- Paragraph 141 - 150 is very speculative. Please, provide references or data/information that supports authors claim.

We agree; this part was not proven. To clarify the Manuscript and make it more unified, we have removed the part describing the xerogels formed on different surfaces.

- The conclusions Section is a mix of discussion and conclusions. Please clarify these paragraph and provide clear conclusions of this study. References should be avoided to highlight just only the authors´ findings.

Thank you for the note. We have revised and thinned out the Conclusions section, but some references are still present in the text, as they are necessary for the reasoning.

- This whole paper needs to be submitted to English language review by a native speaker or a manuscript editing service.

Sorry for bad English. We have completely revised and reassembled the Manuscript and proofread the language.

Round  2

Reviewer 1 Report

The authors did not reply to my comments accurately. The altered title in the reply was “The Influence of the Dispersing Intensity on the Gel-Sol Transition in Aqueous Cellulose Suspensions” , however, the revised manuscript was Twisting of Fibers Balancing the Gel-Sol Transition in Cellulose Aqueous Suspensions.  While I have no objections to the presentation of the results it is up to the Editor to decide whether the work is of significant importance for Polymers journal.

Reviewer 2 Report

The authors addressed all the comments and the paper has been significantly improved.